# Discovery of Oleaginous Yeast from Mountain Forest Soil in Thailand

**DOI:** 10.3390/jof8101100

**Published:** 2022-10-18

**Authors:** Sirawich Sapsirisuk, Pirapan Polburee, Wanlapa Lorliam, Savitree Limtong

**Affiliations:** 1Department of Microbiology, Faculty of Science, Srinakharinwirot University, Bangkok 10110, Thailand; 2Department of Microbiology, Faculty of Science, Kasetsart University, Bangkok 10900, Thailand; 3Biodiversity Center, Kasetsart University, Bangkok 10900, Thailand; 4Academy of Science, The Royal Society of Thailand, Bangkok 10300, Thailand

**Keywords:** oleaginous yeast, *Lipomyces*, lipid production, soil, *Papiliotrema*, screening

## Abstract

As an interesting alternative microbial platform for the sustainable synthesis of oleochemical building blocks and biofuels, oleaginous yeasts are increasing in both quantity and diversity. In this study, oleaginous yeast species from northern Thailand were discovered to add to the topology. A total of 127 yeast strains were isolated from 22 forest soil samples collected from mountainous areas. They were identified by an analysis of the D1/D2 domain of the large subunit rRNA (LSU rRNA) gene sequences to be 13 species. The most frequently isolated species were *Lipomyces tetrasporus* and *Lipomyces starkeyi*. Based on the cellular lipid content determination, 78 strains of ten yeast species, and two potential new yeast that which accumulated over 20% of dry biomass, were found to be oleaginous yeast strains. Among the oleaginous species detected, *Papiliotrema terrestris* and *Papiliotrema flavescens* have never been reported as oleaginous yeast before. In addition, none of the species in the genera *Piskurozyma* and *Hannaella* were found to be oleaginous yeast. *L. tetrasporus* SWU-NGP 2-5 accumulated the highest lipid content of 74.26% dry biomass, whereas *Lipomyces mesembrius* SWU-NGP 14-6 revealed the highest lipid quantity at 5.20 ± 0.03 g L^−1^. The fatty acid profiles of the selected oleaginous yeasts varied depending on the strain and suitability for biodiesel production.

## 1. Introduction

The increasing demand for green bio-based alternatives, including bioenergy such as biodiesel and several oil-related biotechnological applications [1], has resulted in a rapid increase in demand for raw materials such as vegetable oils. However, increasing oil requirements could lead to an increase in the price of food crops [2]. They do not appear to be a sustainable fuel source owing to high demand in the food sector [3,4]. Therefore, the exploration of alternative non-edible resources is necessary to meet this requirement. One of the promising oil feedstocks is microbial oil or single cell oil (SCO), which is produced by oleaginous microorganisms [5,6,7].

Oleaginous yeast is able to accumulate cellular lipids at greater than 20% of its biomass [8]. These yeasts represent a promising potential feedstock for biodiesel production due to the fact that the composition of their fatty acids is similar to that of vegetable oils [9]. In general, cells of oleaginous yeasts accumulate high quantities of lipids when cultivated in nitrogen-limited media with a high C/N ratio; that is, the media contain higher amounts of carbon in comparison to nitrogen [10,11]. Only 11% of the 1600 known yeast species have been classified as oleaginous yeast species [12]. These include various species within the 32 genera of the phylum Basidiomycota (e.g., *Cutaneotrichosporon curvatum*, *Rhodosporidiobolus fluvialis*, *Rhodotorula toruloides*, *Rhodotorula glutinis*, and *Sporidiobolus ruineniae*), and 27 genera of the phylum Ascomycota (e.g., *Candida tropicalis*, *Candida utilis*, *Kodamaea ohmeri*, *Lipomyces starkeyi*, *Lipomyces lipofer*, and *Yarrowia lipolytica*) [13,14,15,16,17,18,19,20,21,22,23].

Currently, the various *Lipomyces* species are considered to be attractive lipid producers because these species have shown higher lipid accumulation (over 60%) and adaptability to consume a wide range of substrates for production [24,25,26,27]. Most *Lipomyces* species are isolated from soil, as soil is the predominant habitat for yeast species [28,29,30,31]. Soil is considered a reservoir for yeasts in underground environments. Soil yeast communities are taxonomically different from those above ground and have adapted to a wide range of environmental circumstances [32]. Several species are viable sources of yeast oil. Thus, the isolation of yeast from soil could enhance our opportunities to obtain oleaginous yeast strains, and to discover novel and newly found oleaginous yeast species. In Thailand, several newly recorded oleaginous yeast species, such as *Pichia manshurica*, *Rh. fluvialis*, *Rhodotorula sphaerocarpa*, and *Saitozyma podzolica,* have been reported [33,34,35,36]. However, the quantity of lipid accumulation in yeast is strain-dependent because significant differences in lipid content have been observed in different strains of the same yeast species [33,34,37,38]. Therefore, there has been an increasing interest in isolating and identifying additional oleaginous yeast species. A wider selection of candidate species leads to a higher possibility of discovering yeast strains that have advantageous properties. Hence, the aim of this study is to identify additional oleaginous yeast species with better lipid-producing abilities, by isolating yeast samples from forest soil in the mountains in northern Thailand and evaluating their lipid accumulation quality.

## 2. Materials and Methods

### 2.1. Yeast Isolation

Soil samples were collected from 22 sites in forests in the mountains in Chiang Rai province, Thailand. The samples were taken at a soil depth of 5–10 cm. One gram of soil was sprinkled onto a nitrogen–depleted agar medium (NDM-agar) plate (20 g L^−1^ glucose, 20 g L^−1^ agar, 0.85 g L^−1^ KH_2_PO_4_, 0.15 g L^−1^ K_2_HPO_4_, 0.5 g L^−1^ MgSO_4_·7H_2_O, 0.1 g L^−1^ NaCl, 0.1 g L^−1^ CaCl_2_·6H_2_O, 0.5 mg L^−1^ H_3_BO_3_, 0.04 mg L^−1^ CuSO_4_·4H_2_O, 0.1 mg L^−1^ KI, 0.2 mg L^−1^ FeCl_3_·6H_2_O, 0.4 mg L^−1^ MnSO_4_·H_2_O, 0.2 mg L^−1^ NaMoO_4_·2H_2_O, 0.4 mg L^−1^ ZnSO_4_·7H_2_O and 0.25 g L^−1^ chloramphenicol) [30] and a nitrogen-depleted gellan gum (NDM-gellan gum) medium plate, which substituted agar with 7.2 g L^−1^ gellan gum. The plates were incubated at room temperature (~25–30 °C) for 2–3 weeks until yeast colonies appeared. Colonies of yeast were isolated and streaked onto a yeast extract malt extract (YM) agar medium (10 g L^−1^ glucose, 15 g L^−1^ agar, 5 g L^−1^ peptone, 3 g L^−1^ yeast extract, 3 g L^−1^ malt extract, pH 5.6). These purified yeast cultures were then maintained on the YM agar slant at 4 °C for further study. The frequency of occurrence (FO%) was calculated as the number of samples in which a particular species was observed divided by the total number of samples.

### 2.2. Identification of Yeast

The D1/D2 domain of the LSU rRNA gene sequence was used to identify the isolated yeast strains. The extraction of genomic DNA was carried out by lysis of yeast cells in 200 µL lysis buffer (100 mM Tris, 30 mM EDTA, 0.5% SDS, pH 8) at 100 °C for 15 min. The cell lysate was mixed with 200 µL 2.5 M potassium acetate (pH 7.5) on ice. After being centrifuged at 13,000 rpm, 4 °C for 5 min, the supernatant was harvested and mixed with 400 µL cold isopropanol. The DNA precipitate was rinsed twice with 200 µL 70% (*v*/*v*) ethanol by centrifugation (14,500 rpm, 4 °C for 5 min) and eluted by 20–50 µL TE buffer. The extracted DNA was used as a template for amplification by PCR (PCRmax Alpha Cycle 1, PCRmax, Staffordshire, United Kingdom). The D1/D2 domain of the LSU rRNA gene was amplified from the genomic DNA using the primer sets NL1 (5′-GCATATCAATAAGCGGAGGAAAAG-3′) and NL4 (5′-GGTCCGTGTTTCAAGACGG-3′) [39]. The PCR product was then checked by agarose gel electrophoresis and purified using a Gel/PCR Purification Kit (Favorgen Biotech, Pingtung, Taiwan). The purified product was submitted to Macrogen Inc. (Seoul, Korea) for sequencing using primers NL1 and NL4. MEGA-X software was used to construct a phylogenetic tree [40], with the bootstrap consensus tree inferred from 1000 replicates. Evolutionary distances were computed using the maximum composite likelihood method.

Analysis of the D1/D2 domain is currently commonly used to identify the great majority of yeast species, and nearly all recognized ascomycete yeasts and basidiomycete yeasts. One advantage of using the D1/D2 domain for yeast identification is the large library of yeast species available [34,36]. The Identification of ascomycete yeast species was established by the generally recognized guideline that strains indicating more than 1% nucleotide substitutions in the ca. 600–500 nucleotides of the D1/D2 domain are expected to be different species, while strains with 0–3 nucleotide differences are either conspecific or sister species [39]. The identification of basidiomycete yeast species followed the suggestion of Fell et al. [41] that the sequence of the D1/D2 domain of strains in this species are identical, and strains with two or more nucleotide divergences in this domain represent different species. However, the PCR amplification condition and technical differences between laboratories have an impact on reproducibility and may influence the results of the sequence analysis of each strain [42].

### 2.3. Screening for Oleaginous Yeast Strains

To screen for oleaginous yeast strains, all the yeast strains were determined for their lipid accumulation in a limited nitrogen source condition using 2G2M medium containing 20 g L^−1^ glucose and 20 g L^−1^ malt extract, which was modified from the medium used by Yamazaki et al. [43]. The cultivation of yeast in nitrogen-limited medium has been reported to promote lipid accumulation of the oleaginous yeast strains [10,11,33,36,44]. To prepare an inoculum, one loopful of isolated yeast strain was inoculated in 50 mL of YM broth in a 250 mL Erlenmeyer flask and incubated on a rotary shaker (ZWYR-211C Labwit, Victoria, Australia) at 30 °C, 150 rpm for 24 h. Cells were collected by centrifugation (3700 Kubota, Tokyo Japan) at 4 °C, 5000 rpm, 10 min, washed with sterile normal saline solution, and adjusted to OD 1 at 600 nm. The inoculum was transferred into 50 mL of 2G2M medium in a 250 mL Erlenmeyer flask and incubated on a rotary shaker at 30 °C, 150 rpm for 7 days. Cells were harvested by centrifugation (4 °C, 5000 rpm, 10 min) and analyzed for biomass and lipid content.

### 2.4. Analytical Methods

#### 2.4.1. Biomass Analysis

Culture broth was harvested, and cells were separated using centrifugation at 8500× *g* for 10 min, washed with distilled water, and dried at 100 °C until constant weight. The biomass was measured gravimetrically.

#### 2.4.2. Lipid Content and Fatty Acid Analyses

For lipid extraction, the method of Bligh and Dyer [45] was used. Yeast cells were suspended in 3.75 mL of a mixture of chloroform and methanol (2:1, *V*/*V* ratio), ultrasonicated at 25 kHz for 10 min, and centrifuged at 4 °C, 3500 rpm for 20 min. The supernatant was collected and dried by evaporation. Lipid quantity was analyzed by the gravimetric method. Fatty acids were converted to fatty acid methyl esters according to the method of Holub et al. [46]. To determine the composition of fatty acids, GC analysis (GC14-A, Shimadzu, Kyoto, Japan) equipped with a capillary column containing a silica megabore column (30 m × 0.52 mm × 1 m, Durabond 225, J & W Scientific, Folsom, CA, USA) was used. The operating conditions were as follows: 10 mL/min of helium carrier gas, 40 mL/min of nitrogen carrier gas, a 210 °C column temperature, and a 250 °C injection temperature and detection temperature [46]. Fatty acid retention times were compared to those of standard fatty acid mixtures.

Cellular lipid bodies of the selected oleaginous yeast were qualitatively evaluated by staining cells with Nile red following the method of Kimura et al. (2004) [40], and evaluated with a fluorescence microscope (Olympus BX51, Tokyo, Japan).

### 2.5. Statistical Analysis

Statistical analysis was performed using SPSS statistical package version 25.0 (IBM SPSS Inc., New York, NY, USA). The data were analyzed using a one-way analysis of variance (ANOVA) followed by Tukey’s test to calculate significant differences in treatment means, and the least significant difference (*p* < 0.05) was considered statistically significant.

## 3. Results and Discussion

### 3.1. Isolation and Identification of Yeasts

All the yeast strains were isolated from 22 soil samples collected from forests on the mountains in Chiang Rai province in northern Thailand. The average pH of the mountain forest soil was determined to be neutral (6.5–7.0). The soil humidity ranged from 15 to 45%, while the temperature ranged from 15 to 21 °C (Appendix A). In total, 127 yeast strains were isolated from the 22 soil samples. They consisted of 56 strains isolated using NDM-agar and 71 strains obtained using NDM-gellan gum. All yeast strains have been maintained at the Department of Microbiology, Faculty of Science, Srinakharinwirot University, and the information of strains has been deposited in the database of the research project FF(KU)18.64. Based on the sequence analysis of the D1/D2 domain of the LSU rRNA gene, the isolated strains were determined to belong to 11 described yeast species and two potentially new species. From 127 yeast strains, 97 strains (76.3%) were identified as five species in the phylum *Ascomycota*, subphylum *Saccharomycotina*. The remaining 31 strains (24.4%) were identified to be eight species in the phylum Basidiomycota in the following subphylums: two species in the subphylum *Pucciniomycotina*, and four species, along with two potential new species, in the subphylum *Agaricomycotina*. The 11 described species were *Cyberlindnera saturnus*, *Lipomyces mesembrius*, *Lipomyces starkeyi*, *Lipomyces tetrasporus*, *Meyerozyma guilliermondii*, *Rhodotorula mucilaginosa*, *Cystobasidium slooffiae*, *Naganishia diffluens*, *Papiliotrema flavescens*, *Papiliotrema terrestris* and *Saitozyma podzolica.* The two undescribed basidiomycete species were closely related to *Piskurozyma taiwanensis* and *Hannaella oryzae* (Table 1).

Yeasts are common among the many kinds of soil microorganisms. A few autochthonous soil yeast species complete their entire life cycle in the soil. The best known of these are the *Lipomyces* species, which can be isolated only from mineral soil horizons [47]. However, the most abundant yeast in soils are allochthonous species that originate from other sources, typically plants and plant residues [33,48]. Based on the results of this study, the most frequently isolated species was *L. tetrasporus* (37.8% FO), followed by *L. starkeyi* (24.4% FO), *L. mesembrius* (7.9% FO), *P. flavescens* (7.9% FO), *S. podzolica* (6.3% FO), and *Cyb. saturnus* (5.5% FO), while the remaining species produced 1–4 strains in total (Table 1).

This revealed that the main habitat of the three *Lipomyces* species, including *L. mesembrius*, *L. starkeyi*, and *L. tetrasporus*, was soil. Several anamorphic basidiomycete species, *S. podzolica*, *Solicoccozyma terricola*, *Apiotrichum* spp., and *Rhodotorula* spp., were isolated from various soils, indicating that soil is the ecological niche of these yeasts [49,50,51]. Likewise, *P. flavescens* and *P. terrestris*, which were detected in this study, were previously reported to isolate from forest soil and phyllosphere [51,52]. Previous studies have shown that *M. guilliermondii*, *N. diffluens* and *S. podzolica* can also be isolated from soil, while *R. mucilaginosa* and *P. flavescens* were reported to isolate from decayed materials and water in the mangrove forests of Thailand [34,38,53]. *Cyb. saturnus*, a xylitol producing yeast species, has been reported in soils in Thailand [53].

Interestingly, *C. slooffiae* has been reported as an exoelectrogenic yeast strain that has been isolated from activated sludge [54]. The survival of these autochthonous soil yeasts in this environment has been attributed to several characteristics. Most of these yeasts have a broad range of metabolic activities that allow them to assimilate mold- and prokaryote-generated hydrolytic plant products and create exopolymeric capsules, which may help them survive in nutrient-poor environments [32]. *Lipomyces* and *Cyb. saturnus* strains are able to produce resistant spores, whereas *Lipomyces*, *Cryptococcus*, and *Rhodotorula* produce exopolymeric capsules [32].

Moreover, the three strains including SWU-NATP 4-12, SWU-NGP 14-3-2, and SWU-NGP 14-3-4 were identified to be potential new yeast species closest to *Piskurozyma taiwanensis* CBS 9813^T^ with 96.5–96.8% identity (18–20 nucleotide substitutions of 517–596 nucleotides). The two strains, namely SWU-NAPS 5-1 and SWU-YGP 11-1, were closest to *Hannaella oryzae* CBS 7194 with 98.0–98.9% identity (12 nucleotide substitutions of 501–597 nucleotides) (Figure 1 and Figure 2). The potential new species isolated in this study need to be further investigated and characterized based on polyphasic taxonomy to be proposed as new yeast species. It should be noted that the two potential new yeast species found in this study were isolated by both NDM-agar and NDM-gellan gum medium. However, using the latter medium proved more beneficial for their isolation.

In this study, the media solidified with gellan gum or agar were used to isolate yeasts and succeeded in isolating the *Lipomyces* species with a 70.1% frequency of occurrence (7.9% for *L. mesembrius,* 24.4% for *L. starkeyi* and 37.8% for *L. tetrasporus*). Gellan gum can be used as an alternative to agar for microbiological media and plant tissue culture media [55]. We found that the NDM-gellan gum medium had the water-like transparency of the gels, which made it easy to pick up colonies from soil. The challenge in isolating yeast strains from the soil was the presence of fungi in the soil, as this could make isolation difficult. On NDM-gellan gum medium, the growth of fungal contamination is slower than on NDM-agar. Zhang et al. [56] demonstrated that the GYG09 (glucose-YNB medium containing 0.9% low acyl gellan gum) medium was suitable for growing *Saccharomyces*, and was superior to the glucose-YNB medium containing 20% (GYA20) agar in terms of higher clarity and lower dosage of the gelling agent when the surface plating method was used. In addition, the use of gellan gum instead of agar was more effective in increasing capturability and cultivating phylogenetically novel microorganisms from the sediment [57]. The isolation techniques by using media solidified with gellan gum have been reported to isolate other groups of microorganisms, such as ammonia-oxidizing bacteria, from soil [58,59,60].

### 3.2. Screening for Oleaginous Yeast Strains

To screen for oleaginous yeast strains, all 127 isolated yeast strains were evaluated for lipid and biomass production by shaking flask cultivation in 2G2M broth for 7 days (Appendix A). The results showed that 78 yeast strains (61.4%) accumulated lipid content higher than 20% of dry biomass, indicating that they are oleaginous yeast strains. Among these, 26 strains (33.3%) accumulated lipid content at 20–25% of dry biomass, 19 strains (24.3%) accumulated lipid content at 25–30% of dry biomass, and 18 strains (23.1%) revealed lipid contents of 30–40% of dry biomass (Figure 3). This study revealed higher rates of oleaginous yeast isolation (61.4%) than those obtained in previous research, where only 5% of the isolated strains were oleaginous strains [33,61].

Fifteen strains (19.2%) accumulated lipid content higher than 40% of dry biomass and belonged to *Cyb**. saturnus*, *L. mesembrius, L. starkeyi*, *L. tetrasporus*, *M. guilliermondii*, *P. flavescens*, and *S**.podzolica* (Table 2). Moreover, the two undescribed oleaginous yeast strains, *Hannaella* sp. SWU-YGP 11-1 and *Piskurozyma* sp. SWU-NATP 4-12, showed the ability to produce lipid and biomass concentrations of 0.36 ± 0.08 g L^−1^ and 1.70 ± 0.15 g L^−1^, and 0.74 ± 0.04 g L^−1^ and 2.92 ± 0.24 g L^−1^, respectively (Table 3). Importantly, *L. tetrasporus* SWU-NGP 2-5 had the highest lipid content of 74.26% ± 6.30 of dry biomass, while *L. mesembrius* SWU-NGP 14-6 produced the highest lipid quantity of 5.20 ± 0.03 g L ^−1^. In this study, the maximum lipid accumulation of *L. tetrasporus* SWU-NGP 2-5 in glucose containing medium under non-optimal conditions was 74.26% of dry biomass. Interestingly, *L. tetrasporus* SWU-NGP 2-5 revealed higher lipid content than that reported for *L. tetrasporus* (20.0–63.7% of dry biomass) in a previous study [12]. The lipid bodies in the Nile red staining cells of *L. tetrasporus* SWU-NGP 2-5 and *L. mesembrius* SWU-NGP 14-6 were represented as a yellow gold color under a fluorescence microscope (Figure 4).

Several studies reported the lipid production by *L. tetrasporus* from lignocellulosic hydrolyzates. Xue et al. [62] reported that *L. tetrasporus* NRRL Y-11562 was reported to grow better in ammonia fiber expansion-pretreated cornstover hydrolyzate (AFEX CSH) but produced a smaller lipid yield when compared to synthetic media. Minimal washing of AFEX-CS improved the lipid yield and lipid quantity to 0.10 g g^−1^ consumed sugar and 10.7 g L^−1^, respectively. Similar to Slininger et al., [63] reported that *L. tetrasporus*, *Lipomyces kononenkoae* and *Saitoella coloradoensis* produced lipid in the range of 25–30 g L^−1^ from ammonia fiber expansion-pretreated cornstover hydrolyzate (AFEX CSH) and the harsher switchgrass hydrolyzate (SGH) under optimum conditions using a two-stage process. *L. tetrasporus* Y-11562 produced lipid content of 16.3–20.8 g L^−1^ with cultivation in a mixture of glucose and xylose at a 1:1 ratio [63]. Caporusso et al. [64] reported that *L. tetrasporus* Li-0407 accumulated lipid content at 47% of dry biomass when cultivated in undetoxified cardoon hydrolysate.

*L. mesembrius* SWU-NGP 14-6 accumulated lipid content up to 57.1% of dry biomass, with a highest lipid concentration of 5.20 ± 0.03 g L^−1^ and a highest biomass concentration of 9.13 ± 0.36 g L ^−1^. Few articles on lipid production by *L. mesembrius* have been published, but it has not been previously reported as a contender for lipid production. However, this is the first report on lipid accumulation of greater than 50% of dry biomass by *L. mesembrius*. Juanssilfero et al. [65] reported the cultivation of *L. mesembrius* in a nitrogen-limited mineral medium containing a mixture of glucose and xylose as carbon sources and reported a high lipid accumulation of 41.89 ± 1.94% of dry biomass. Currently, well-known oleaginous yeast species such as *Y. lipolytica*, *R. torulodies*, *L. starkeyi*, and *R. glutinis* are of interest as potential lipid producers due to their high lipid content of over 50% of dry biomass, ability to grow on a wide range of substrates, and good growth performance [12,66,67,68]. Therefore, in the present study, *L. tetrasporus* SWU-NGP 2-5 and *L. mesembrius* SWU-NGP 14-6 were promising candidates for lipid production.

### 3.3. Discovery of Additional Oleaginous Yeast Species

In this study, 78 oleaginous yeast strains were found to belong to ten yeast species, including *L. tetrasporus*, *L. starkeyi*, *L. mesembrius*, *Cyb. Saturnus*, *M. guilliermondii*, *S. podzolica*, *P. flavescens*, and *P. terrestris*, along with two potential new yeast species: *Piskurozyma* sp. SWU-NATP 4-12 and *Hannaella* sp. SWU-YGP 11-1 (Figure 5). *C. slooffiae* (1 strain), *N. diffluens* (1 strain), and *R. mucilaginosa* (1 strain) accumulated lipid content lower than 20% of dry biomass (Figure 5); hence, they were not included as oleaginous yeast species in this study.

The lipid content varied significantly among the species evaluated; furthermore, there was variation in lipid content when comparing multiple strains of the same species. The highest number of oleaginous yeast strains present in this study were in *L. tetrasporus.* Of the 48 identified strains of this species, 28 strains (58.3%) were classified as oleaginous strains. Of the 31 strains of *L. starkeyi*, 20 strains (64.5%) were classified as oleaginous strains. Of the 10 strains of *L. mesembrius* isolated, nine were classified as oleaginous strains (90.0%), and seven of the ten isolated strains (87%) of *S. podzolica* were oleaginous strains. These results demonstrate that strains in the same species accumulate lipid content in the range of 20–70%, indicating that lipid accumulation is strain-dependent, not species- or genus-dependent [33].

This strain-dependent lipid accumulation is also evident among stains of *Y. lipolytica*. Ngamsirisomsakul et al. [37] reported that the lipid content of *Y. lipolytica* strains cultivated in medium containing xylose as a sole carbon source of the strain TISTR 5212 was 14% of dry biomass, whereas the strain TISTR 5054 accumulated lipid of 4% of dry biomass. Moreover, the ability to accumulate lipid was different in *Trichosporon*
*cutaneum* strains. Three strains, including TISTR 5040, TISTR 5083, and TISTR 5133, accumulated lipid at approximately 11% of dry biomass; in contrast, strains TISTR 5084 and TISTR 5112 accumulated higher lipid levels of approximately 19% and 18% of dry biomass, respectively [37].

In this study, *L. tetrasporus*, *L. starkeyi* and *L. mesembrius* produced lipid content at 20.0–74.3% of dry biomass. Among these species, *L. starkeyi* is regarded as the species with the highest biotechnology value due to its highly attainable lipid content, inhibitor tolerance, and its ability to utilize mixed carbon sources such as lignocellulosic hydrolysate [12,63,64]. Caporusso et al. [64] and Dien et al. [68] have reported that *L. tetrasporus* Li-0407 and Y-11562 produced high lipid contents of 41.1%–68.6% of dry biomass when grown in optimal conditions in a nitrogen-limited medium containing glucose and diluted non-detoxified carbon hydrolysate. *S. podzolica* (syn. *Cryptococcus podzolicus*) [16,69,70,71] and *M. guilliermondii* [72,73] have been reported to produce lipid content when cultivated in agricultural waste and crude glycerol, while *Cyb. saturnus* has been reported to grow in volatile fatty acid for lipid production [74].

Strains of *P. terrestris* and *P. flavescens* were found to be oleaginous strains only in this study. Therefore, this is the first report in which these two species can be classified as oleaginous yeast species. In the genus *Papiliotrema,* only *Papiliotrema laurentii* has been reported as oleaginous; the strains UFV-1 and UFV-2 showed a strong lipid production and high lipid content of up to 63% [75]. Of the five strains of potential new yeast species, only *Hannaella* sp. SWU-YGP 11-1 and *Piskurozyma* sp. SWU-NATP 4-12 were found to be oleaginous strains (Table 3). Species in the genus *Hannaella* have been isolated from various sources such as soil, plants, and water [75,76]. Only *Hannaella* aff. *zeae*, has been identified as oleaginous in a single publication [76]. Species in the genus *Piskurozyma* have not previously been reported to be classified as oleaginous yeast; therefore, this is the first report in which a member of the genus *Piskurozyma* has accumulated a high enough lipid yield to be classified as oleaginous. Thus, this study has been able to add additional oleaginous yeast species.

### 3.4. Fatty Acid Composition Profiles

On average, oleaginous yeast strains can accumulate lipids to a level corresponding to 40% of dry biomass [77]. However, under nutrient-limiting conditions, some oleaginous yeast strains have shown the capacity to accumulate lipid content greater than 70% of dry biomass [20,77,78,79]. These lipids are predominantly composed of triacylglycerols (80–90%), which usually contain long-chain fatty acids of 16 or 18 carbon atoms [7,80,81,82,83].

The fatty acid composition profiles of lipids derived from selected oleaginous yeast in this study are summarized in Table 4. Linoleic acid (C18:2) and palmitic acid (C16:0) are the major fatty acids of the lipids produced by most oleaginous yeast strains, followed by oleic acid (C18:1) and stearic acid (C18:0). This suggests that yeast lipids are appropriate as potential alternative feedstocks for biodiesel production because they have similar fatty acid compositions to those of plant oils [61]. The fatty acid profile showed that all strains of the same species have the same major fatty acids (C18:1 and C16:0), while some other fatty acids show a slight difference between the strains. However, the fatty acid profiles of yeast lipids vary by type of growth phase, culture medium components (carbon source and nitrogen source), and culture conditions (temperature, pH, inoculum size, cultivation time, and dissolved oxygen level) [84,85,86].

## 4. Conclusions

The results obtained in this study show a variety of yeast species isolated from forest soil collected from three mountains in northern Thailand. Thirteen species belonging to both phyla *Ascomycota* and *Basidiomycota* were obtained. Of the 127 strains collected, 78 strains (61.42% of isolated strains) were classified as oleaginous strains due to their lipid accumulation. The oleaginous yeast strains were identified as *L. tetrasporus, L. starkeyi, L. mesembrius, Cyb. saturnus M. guilliermondii, S. podzolica*, *P. flavescens*, and *P. terrestris*, along with two potential new yeast species, *Piskurozyma* sp. and *Hannaella* sp. Only *P. flavescens*, *P. terrestris*, *Piskurozyma* sp., and *Hannaella* sp. were found to be oleaginous yeasts in this study. *L. tetrasporus* SWU-NGP 2-5 revealed the highest lipid content, while *L. mesembrius* SWU-NGP 14-6 produced the highest lipid quality. The fatty acid compositions of the oleaginous yeast strains were similar to those of vegetable oils; thus, yeast lipids have the potential to be used as an alternative feedstock for biodiesel and oleochemical production.

## Figures and Tables

**Figure 1 jof-08-01100-f001:**
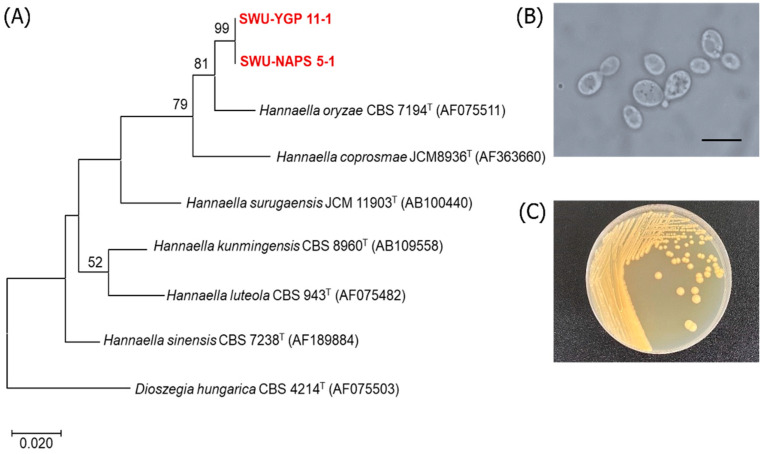
Phylogenetic tree (**A**) showing the positions of the strains SWU-YGP 11-1 and SWU-NAPS 5-1 and their related species constructed by the maximum-likelihood method based on the D1/D2 of the LSU rRNA gene sequences. *Dioszegia hungarica* CBS 4214^T^ was used as an outgroup. The numerals represent the percentages from 1000 replicate bootstrap resampling (a frequency of less than 50% is not shown). The cells (**B**) and colony (**C**) morphology of SWU-YGP 11-1 grown on YM agar for 48 h. The bar represents as 10 µm.

**Figure 2 jof-08-01100-f002:**
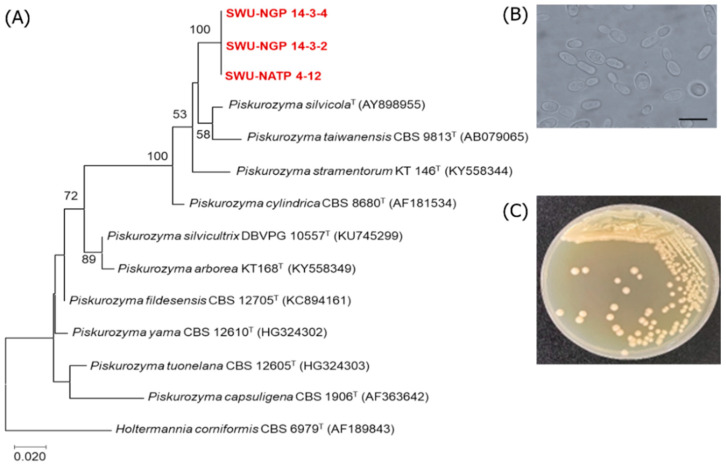
Phylogenetic tree (**A**) showing the positions of the strains SWU-NGP 14-3-2, SWU-NGP 14-3-4, SWU-NGTP 4-12, and their related species constructed by the maximum-likelihood method based on the D1/D2 of the LSU rRNA gene sequences. *Holtermannia corniformis* CBS 6979^T^ was used as an outgroup. The numerals represent the percentages from 1000 replicate bootstrap resampling (a frequency of less than 50% is not shown). The cells (**B**) and colony (**C**) morphology of SWU-YGP 11-1 grown on YM agar for 48 h. The bar represents as 10 µm.

**Figure 3 jof-08-01100-f003:**
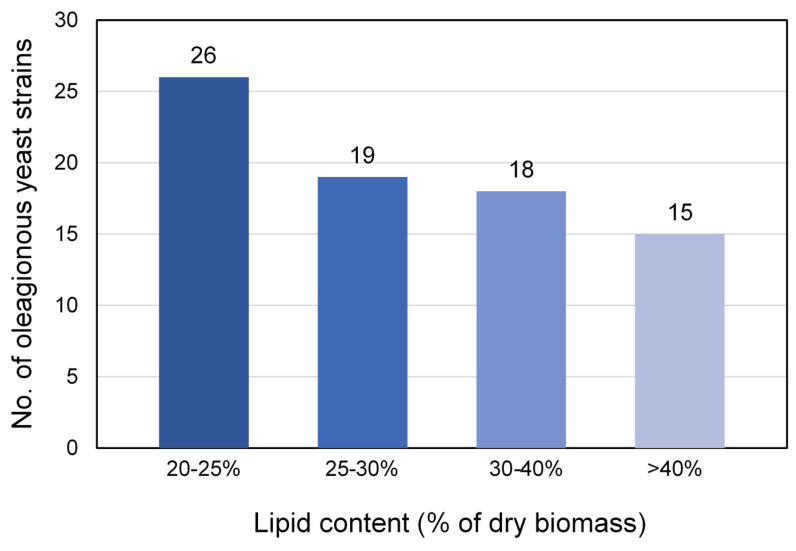
The number of oleaginous yeasts in this study. The lipid accumulation in the rage of 20–25% of dry biomass (■), 25–30% of dry biomass (■), 30–40% of dry biomass (■) and >40% of dry biomass (■). X-axis is the percentage of lipid content and Y-axis is number of yeast strains.

**Figure 4 jof-08-01100-f004:**
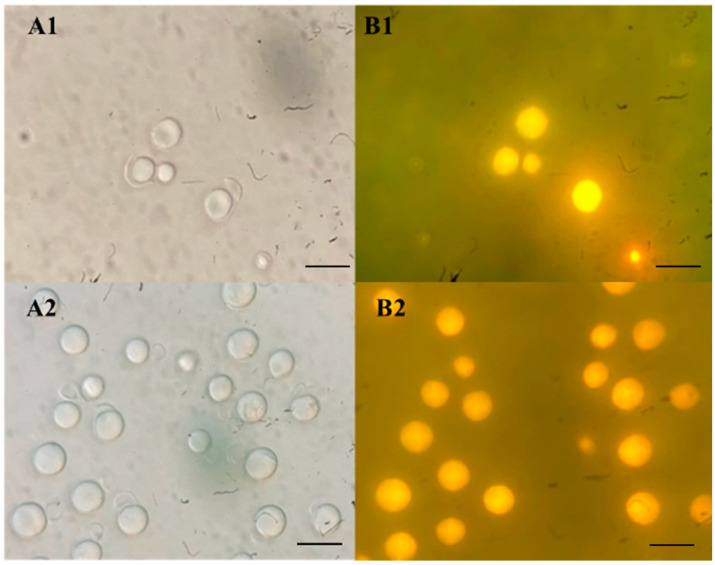
Photomicrographs of cell under light microscope (**A1**,**A2**) and lipid bodies by Nile red staining under fluorescence microscope (**B1**,**B2**) of *Lipomyces tetrasporus* SWU-NGP 2-5 (**A1**,**B1**) and *Lipomyces mesembrius* SWU-NGP 14-6 (**A2**,**B2**) grown in 2G2M broth for 7 days. The bar represents as 10 µm.

**Figure 5 jof-08-01100-f005:**
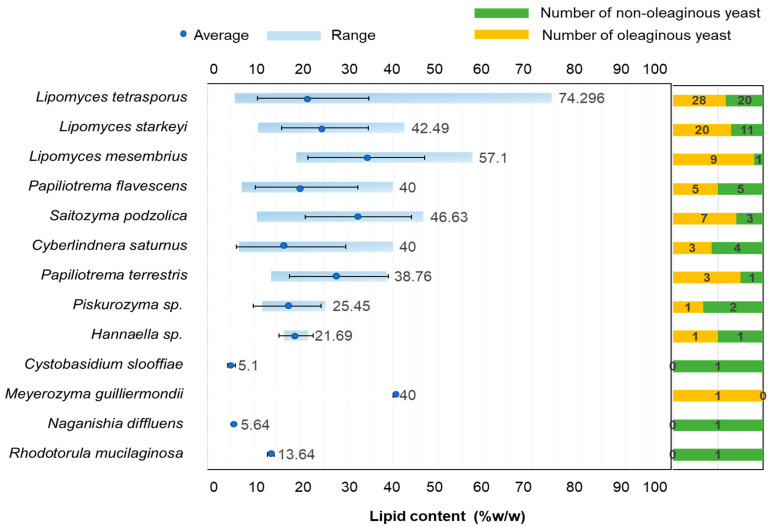
Summary of 13 species isolated from forest soil on the mountains in Thailand. The range of the highest lipid content including average and standard deviation (blue bar), as well as the number of oleaginous yeasts (yellow bar) and the number of non-oleaginous yeasts (green bar), are shown.

**Table 1 jof-08-01100-t001:** Frequency of occurrence of each yeast species isolated from mountain forest soil.

Phylum and Subphylum	Species	Number of Strains	FO (%) ^a^
*Ascomycota* (97 strains)			
*Saccharomycotina*	*Cyberlindnera saturnus*	7	5.5
	*Lipomyces mesembrius*	10	7.9
	*Lipomyces starkeyi*	31	24.4
	*Lipomyces tetrasporus*	48	37.8
	*Meyerozyma guilliermondii*	1	0.8
*Basidiomycota* (31 strains)			
*Pucciniomycotina*	*Rhodotorula mucilaginosa*	1	0.8
	*Cystobasidium slooffiae*	1	0.8
*Agaricomycotina*	*Naganishia diffluens*	1	0.8
	*Papiliotrema flavescens*	10	7.9
	*Papiliotrema terrestris*	4	3.2
	*Saitozyma podzolica*	8	6.3
	*Piskurozyma* sp.	3	2.4
	*Hannaella* sp.	2	1.6
	Total number of strains	127	100.0

^a^ FO: Frequency of occurrence (%) = number of samples where a particular species was observed as a proportion of the total number of strains. Note: All yeast strains have been maintained at the Department of Microbiology, Faculty of Science, Srinakharinwirot University, and the information of strains have been deposited in the database of the research project FF(KU)18.64. The sequences of the D1/D2 domain of the LSU rRNA gene are shown in Appendix A.

**Table 2 jof-08-01100-t002:** Lipid accumulation of oleaginous yeast strains that accumulated lipids higher than 40% of dry biomass cultivated in 2G2M broth containing 20 g L^−1^ glucose on a rotary shaker at 150 rpm and 30 °C for 7 days.

Yeast Strain	Biomass (g L^−1^)	Lipid (g L^−1^)	Lipid Content (%)
*L. tetrasporus* SWU-NGP 2-5	1.49 ± 0.05 ^g^	1.10 ± 0.06 ^gh^	74.26 ± 6.30 ^a^
*L. tetrasporus* SWU-NAP 4-1	4.50 ± 0.08 ^ef^	2.57 ± 0.04 ^d^	57.10 ± 1.82 ^b^
*L. mesembrius* SWU-NGP 14-6	9.13 ± 0.36 ^a^	5.20 ± 0.03 ^a^	57.10 ± 1.89 ^b^
*L. mesembrius* SWU-NGP 6-4	7.83 ± 0.17 ^b^	3.98 ± 0.02 ^b^	50.80 ± 0.80 ^b^
*L. tetrasporus* SWU-NGP 5-8-1	9.07 ± 0.16 ^a^	4.27 ± 0.01 ^b^	47.06 ± 0.66 ^bcd^
*S. podzolica* SWU-NAP 5-4-2	3.88 ± 0.19 ^ef^	1.81 ± 0.19 ^ef^	46.63 ± 2.51 ^cde^
*L. starkeyi* SWU-NGP 14-3-1	3.53 ± 0.56 ^f^	1.47 ± 0.07 ^fg^	42.49 ± 4.65 ^cde^
*L. starkeyi* SWU-NGTP 4-5	5.92 ± 0.14 ^cd^	2.45 ± 0.22 ^d^	41.56 ± 4.67 ^de^
*L. starkeyi* SWU-NGP 5-7	8.39 ± 0.85 ^ab^	3.35 ± 0.00 ^c^	40.30 ± 4.11 ^e^
*Cyb. saturnus* SWU-NGTP 5-3-3	1.00 ± 0.04 ^gh^	0.4 ± 0.014 ^ij^	40.00 ± 0.00 ^e^
*S. podzolica* SWU-NGP 5-3-8	6.59 ± 0.89 ^c^	2.63 ± 0.35 ^d^	40.00 ± 0.00 ^e^
*P. flavescens* SWU-NGTP 4-1	1.87 ± 0.24 ^g^	0.74 ± 0.10 ^hi^	40.00 ± 0.00 ^e^
*S. podzolica* SWU-NGP 14-2-2	4.99 ± 0.19 ^de^	1.99 ± 0.07 ^e^	40.00 ± 0.00 ^e^
*M. guilliermondii* SWU-NATP 2-4	0.25 ± 0.04 ^h^	0.10 ± 0.02 ^j^	40.00 ± 0.00 ^e^
*P. flavescens* SWU-NATP 3-3	1.15 ± 0.06 ^gh^	0.46 ± 0.02 ^ij^	40.00 ± 0.00 ^e^

Data were reported as mean ± standard deviation (*n* = 3). Different letters indicate statistically significant differences (column). One-way ANOVA and Tukey test (*p* < 0.05).

**Table 3 jof-08-01100-t003:** Lipid accumulation of five strains of the potential new yeast species.

Yeast Strain	Closely Related Species (Accession Number)	Similarity (%)	Gaps/Total Nucleotide	Nucleotide Substitution	Biomass (g L^−1^)	Lipid (g L^−1^)	Lipid Content (% of Dry Biomass)
*Hannaella* sp. SWU-YGP 11-1	*Hannaella oryzae* CBS 7194^T^ (AF075511)	98.9	1/597	12	1.70 ± 0.15	0.36 ± 0.08	21.70
*Hannaella* sp. SWU-NAPS 5-1	*Hannaella oryzae* CBS 7194^T^ (AF075511)	98.0	1/501	12	6.41 ± 0.65	1.10 ± 0.15	16.46
*Piskurozyma* sp. SWU-NATP 4-12	*Piskurozyma taiwanensis* CBS 9813^T^ (AF079035)	96.5	2/596	19	2.92 ± 0.24	0.74 ± 0.04	25.45
*Piskurozyma* sp. SWU-NGP 14-3-2	*Piskurozyma taiwanensis* CBS 9813^T^ (AF079035)	96.8	2/517	18	0.53 ± 0.15	0.07 ± 0.01	14.12
*Piskurozyma* sp. SWU-NGP 14-3-4	*Piskurozyma taiwanensis* CBS 9813^T^ (AF079035)	96.8	2/583	19	0.67 ± 0.01	0.08 ± 0.01	11.72

**Table 4 jof-08-01100-t004:** Fatty acid composition of lipids produced by the selected oleaginous yeast strains.

Strains	Relative Content of Fatty Acid (% *w*/*w*)							
	C14:0	C16:0	C16:1	C18:0	C18:1	C18:2	C18:3α	C18:3β
*L. mesembrius* SWU-NAP 3-4	0.55	41.7	1.5	8	0.7	15.7	-	1.6
*L. mesembrius* SWU-NAP 8-4	0.38	33.4	0.1	7.7	1	55.1	0.3	1.9
*L. tetrasporus* SWU-NAP 5-3-1	0.32	41.1	0.6	-	6.8	48.6	0.9	1.6
*L. tetrasporus* SWU-NAP 13-8	0.22	34.5	0.3	11.4	3.2	47.1	0.6	2.6
*P. terrestris* SWU-NGPui 12-9	0.24	19.5	-	17.9	0.8	49.1	-	12.4
*P. terrestris* SWU-NAPui 14-5	-	30.2	-	24.8	13.8	23.4	1.3	6.5
*S. podzolica* SWU-YGP 8-1-2	-	20.3	-	11.6	0.3	63.1	-	4.6
*S. podzolica* SWU-NGP 5-3-2	0.13	19.4	0.2	17.0	0.6	55.8	-	6.9

## Data Availability

Not applicable.

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
