# Peer review of "Discovery of Oleaginous Yeast from Mountain Forest Soil in Thailand"

_jof, 2022, doi:10.3390/jof8101100_

Round 1
Reviewer 1 Report
First, the discovery of novel oleaginous yeast from mountain forest soil in Thailand would be helpful to provide more candidate yeasts for the research and application. Hence, this research is meaningful. Overall, the manuscript is readable and reasonable. More references related to Lipomyces starkeyi can be supplemented since this oleaginous species was considered as a promising one for microbial lipid production. Several related publications have been publications, which can be incorporated by this manuscript to improve its quality.
Second, the presentation of the discovery of novel oleaginous yeast is not attractive enough. One suggestion for the authors is to supplement some experimental figures, such as microscopic images of the novel images with Sudan black staining.
Third, a schematic work flow for discovery of oleaginous yeast from mountain forest soil in Thailand can be supplemented as a Graphical Abstract, which would also make the manuscript much more attractive to the readers.
Author Response
We appreciate you and the reviewers for your precious time in reviewing our manuscript and providing valuable comments. The authors have carefully considered the comments and tried their best to address every one of them. "Please see the attachment."

Reviewer 2 Report
Dear authors,
I consider the manuscript very interesting and of overall high merit, and I think that, after a careful revision, it deserves to be published.
Please find below my major and minor comments, that I consider that will, in an expedite but mandatory way, improve the quality of the manuscript:
Abstract and all the text, generally: english needs to be carefully revised!
1) line 45: many important citations are missing. In particular the following citations are mandatory to be included, but others are missing!
- Ageitos, J.M.; Vallejo, J.A.; Veiga-Crespo, P.; Villa, T.G. Oily yeasts as oleaginous cell factories. Appl. Microbiol. Biotechnol. 2011, 90, 1219–1227.
- Miranda, C.; Bettencourt, S.; Pozdniakova, T.; Pereira, J.; Sampaio, P.; Franco-Duarte, R.; Pais, C. Modified high-throughput Nile red fluorescence assay for the rapid screening of oleaginous yeasts using acetic acid as carbon source. BMC Microbiol. 2020, 20, 60.
- Bettencourt, S; Miranda, C.; Pozdiankova, T.A.; Sampaio, P.; Franco-Duarte, R.; Pais, C. Single Cell Oil Production by Oleaginous. Microorganisms. 2020, 8, 1809.
2) line 57: the strain-dependency of lipid accumulation has only recently been shown and confirmed, so authors need to discuss this fact and add references
3) lines 98-104: please discuss the use of this region to identify yeast species. Please check doi:10.1002/elps.201000640 for a careful consideration about pcr conditions to type yeast species.
Also, Fell et al (doi:10.1099/00207713-50-3-1351), have a nice discussion about D1/D2 domain on basidiomycetous yeasts. These papers need to be cited.
4) section 2.3: authors need to explain the use of this medium, and the source of the experimental design
5) line 141: "the yeast" - which yeast??
6) Lines 174, 176 and others: species abreviations are wrong.
7) Lines 186-195: I don´t consider the evidence to be strong enough for authors to consider the isolates to be new species. Other genomic regions should be considered, and other tests should also be employed (see doi:10.1099/ijsem.0.004531) as example.
Author Response

(The authors gave the same response as above.)

Round 2
Reviewer 2 Report
Authors have answered the majority of the questions, and I consider the manuscript to have improved greatly its overall merit.
I have only two additional minor comments that need to be adressed:
- authors claim that they use taxonomic abbreviations, according to Kurtzman, et al. (2011) and Fell, et al. (2000). However, taxonomic rules have changed since then, and authors should respect them. Please consider the actual Species Taxonomy Nomenclature. For example, Cystobasidium slooffiae, should be C. slooffiae and not Cy. slooffiae. I leave it to editor to decide if authors can leave it as it is, or if they should change it.
- point 3) authors have discussed the use of D1/D2 in a very satisfactory way. Thank you for improving the discussion of this point. However, authors should also include information about the way that the use of different PCR conditions can affect the outcome. Please consider and cite doi:10.1002/elps.201000640, alerting to the fact that results can be influence by this.
Author Response
We appreciated you and the reviewer for your precious time in reviewing our manuscript and providing valuable comments. The authors have carefully considered the comments and tried their best to address them. Please see the attachment.
